# The Active Site of the Enzyme 10-Formyl-THFDH in the Honey Bee *Apis mellifera*—A Key Player in Formic Acid Detoxification

**DOI:** 10.3390/ijms24010354

**Published:** 2022-12-26

**Authors:** Moritz Mating, Ye Zou, Soroush Sharbati, Ralf Einspanier

**Affiliations:** 1Institute of Veterinary Biochemistry, Veterinary Medicine, Freie Universität Berlin, 14163 Berlin, Germany; 2Department of Biochemistry and Molecular Biophysics, Kansas State University, Manhattan, KS 66506, USA

**Keywords:** *Apis mellifera*, 10-formyl-THFDH, mutagenesis, formic acid, detoxification, honey bee

## Abstract

Honey bees are important managed pollinators that fulfill important ecological and economic functions. In recent decades, the obligate ectoparasite *Varroa destructor* severely affected the survival of honey bees, as it weakened them by different means. A common treatment against *V. destructor* is formic acid fumigation, which has been used for decades by beekeepers across the world. This treatment is known to be effective, but many beekeepers report adverse effects of formic acid on bees, which include damage to the brood, worker bee mortality, and queen loss. Little is known about the molecular mechanisms of formic acid detoxification in honey bees. Recently, we reported upregulation of the bee enzyme, 10-formyl-THFDH, under formic acid fumigation. Here, the active site of this enzyme is characterized by an interdisciplinary approach combining homology modeling and protein mutagenesis. In addition, the limitations of the 3D protein structure prediction program AlphaFold2 are shown in regard to docking studies. This study provides a more thorough understanding of the molecular detoxification mechanisms of formic acid in *Apis mellifera*.

## 1. Introduction

Honey bees are known to play a key role in our food production and economy [1,2]. Since the vast majority of plants rely on pollination by bees [3] and crops are the basis for food and feed production, the rapid decline in honey bee colonies in recent decades [4,5] is worrying and of great concern not only to beekeepers but to the public in general. Various factors, owing to malpractices by beekeepers, increased use of pesticides by farmers, the emergence and prevalence of pathogens, and the mite *Varroa destructor* [6], play a key role in colony collapses [7,8]. *V. destructor* as an obligate ectoparasite, which feeds on the fat body and hemolymph of larvae and adult bees [9], directly affects their health. In addition, *V. destructor* is known as a vector of various pathogens, including viruses, such as deformed wing virus and chronic bee paralysis virus [10,11], as well as bacterial pathogens such as *Paenibacillus larvae* [12].

Organic acids such as formic acid are among the most important substances for treatment against *V. destructor* infestation. Due to its low risk of leaving residues in bee products, when applied correctly [13,14], it is licensed for the use in most parts of the world, including the EU, the USA, Canada, and most of Latin America [15,16,17,18,19,20]. Another positive aspect is that, so far, there is no known development of resistance to this treatment. Owing to the many different factors influencing the efficiency of the treatment, such as humidity, temperature, or type and placement of applicator, the treatment with formic acid often includes adverse effects [8,16]. Even though this treatment has been used by beekeepers for decades, its mode of action and the molecular basis of detoxification in honey bees are mainly unknown. Two of our recent studies showed transcriptional upregulation of the mRNA of the enzyme 10-formyl tetrahydrofolate dehydrogenase (10-formyl-THFDH) in honey bees treated with formic acid [21] and the detoxification capability of the recombinantly expressed enzyme [22]. The most recent study also revealed that the enzyme itself is poorly conserved between *Apis mellifera* and the well-described mammalian orthologs. The overall conservation is found to lie between 55% and 70%, whereas the conservation within mammals is found to be higher than 90%. This raises the question of whether the enzyme utilizes the same active sites for the conversion of its substrate.

The ability of 10-formyl-THFDH to completely oxidize 1C units to CO_2_ in an NADP^+^-dependent reaction could remove formic acid from the organism [23]. As reported for mammals, the folate-dependent one-carbon pool (C1) is the most important detoxification pathway of formic acid, catalyzing the conversion of tetrahydrofolate (THF) to 10-formyl-tetrahydrofolate (10-formyl-THF) by a 10-formyltetrahydrofolate synthase [24]. Subsequently, the aforementioned 10-formyltetrahydrofolate dehydrogenase catalyzes the NADP^+^-dependent reaction of 10-formyl-THF to CO_2_ and THF [25,26].

In 2021, the field of protein biochemistry was revolutionized by the AI program AlphaFold2, which allows for atomic close predictions of folded proteins solely based on their primary amino-acid sequence, even if no similar structure is known [27]. AlphaFold2 can be used for different applications, such as the design of expression constructs, de novo protein design, and 3D structure solution. As a result, time-consuming methods for molecular prediction, such as nuclear magnetic resonance (NMR), can be replaced by such machine learning-based approaches. Nevertheless, for complex interactions in and between large macromolecules, such as the interaction between RNA and proteins and in-between proteins, experimental techniques, including but not limited to NMR and X-Ray crystallography, can be applied [28,29]. These laboratory techniques are also the basis for a commonly used approach for structural prediction based on homology modeling. First structures of the target protein are often derived by either SWISS-MODEL [30] or I-TASSER [31]. Thereafter, the 3D model of a target is predicted using the 3D structures of known related proteins, which have been resolved experimentally. These compositions are used to determine the folding and the orientation of sidechains of amino acids. The structural prediction approach is based on evolutionary relationships, in which enzymes that are closely related are expected to have a similar folding [32]. 

Because our enzyme of interest belongs to the group of enzymes utilizing cofactors (NADP^+^), we decided to characterize the interacting sites using an established approach [33]. By combining homology modeling and docking, with mutagenesis of the identified active site, the importance of the main amino acids essential for the enzymatic reaction can be confirmed. Furthermore, we aimed to compare the conservative method with the new AlphaFold2 algorithm in terms of the quality of structural prediction. Lastly, this work pursues the goal of identifying the reactive residues of this detoxifying enzyme in the enzyme of *A. mellifera* to enable future targeted applications for the control of *V. destructor*.

## 2. Results

### 2.1. Homology Modeling, Docking, and Sequence Alignment of Wildtype Enzyme

The consensus sequence (Accession: XP_006563851.2) of 10-formyl-THFDH was used as a basis for the homology modeling with the 10-formyl-THFDH of *Rattus norvegicus* (Accession: NP_071992.2) for structure and side chain orientation, as the structure of the rat ortholog has been solved by crystallography [34].

To examine the conservation of this enzyme within the family of *Apidae* and between mammals and *Apidae*, a multiple-sequence alignment was performed using ClustalΩ [35]. The alignment of the resulting 17 sequences revealed that the active site is very well conserved with only minor exchanges (Appendix A). The key residues and their surrounding regions are conserved with a 100% fit. 

The homology modeling and docking revealed a catalytic subdomain and a NADP^+^-binding subdomain with its NADP^+^-binding site and a substrate entrance tunnel (Figure 1A). The docking also revealed a binding pocket of the substrate 10-formyl-THF with a negatively charged cavity, which is capable of binding the positively charged region of the substrate (Figure 1B,C). Two interacting amino-acid residues were predicted in the enzyme: E673 and C707. Subsequently, both residues were mutagenized to validate their function.

The superimposed structures of the apo-protein and the NADP^+^-binding form revealed a closed loop in the apo-form, which sterically clashes with the substrate (Figure 2A). In the NADP^+^-bound form, the loop opens and reveals the substrate entrance tunnel and the negatively charged cavity. Binding in this pocket allows the substrate to be covalently bound to the C707 which initiates multiple steps to fully oxidize the formyl group. After the formation of the thiohemiacetal, a hydrogen is donated to the NADP^+^ cofactor which results in the formation of a thioester. The following step includes the addition of another hydroxy group, whereafter the E673 donates its electron to accept the hydrogen atom of before mentioned hydroxy group. In the following deacetylation step, a tetrahedral intermediate is formed, which allows the cysteine to break the bond, with the product resulting in the formation of a carboxyl group which is oxidized to CO_2_ (Figure 2B,C).

### 2.2. Homology Modeling and Docking of Mutants

The homology modeling of the mutants was performed as described above. The amino-acid sequences were modified with punctual mutations of the critical residues identified by the docking of the wildtype enzyme with NADP^+^ and the substrate. The following residues were exchanged: C707 to A707 (C707A), E673 to A673 (E673A), and E673 to D673 (E673D). The homology modeling and docking of the mutants with the substrate revealed that no covalent bond could be formed between the alanine mutation of the residue C707 and the substrate formyl group. The residue E673 is expected to activate a water molecule. The E673D exchange, providing the same reactive carboxyl group, is expected to show similar binding capability compared to the wildtype, with the length shortened by one CH_2_ molecule. The alanine exchange is expected to have lower binding capability as the exchange denies the ability to activate the water molecule. Interestingly, a steric clash is seen in all mutants between the substrate and the phenylalanine at position 870 of the enzyme (Appendix A). 

### 2.3. Comparison of AlphaFold2 Model and Homology Models

The consensus sequence (Accession: XP_006563851.2) of 10-formyl-THFDH was subjected to the AlphaFold2 algorithm for structural prediction. The superimposed structures of the AlphaFold2 prediction and the apo-protein (Figure 3A) are both very similar. Overall, both models showed a root-mean-square deviation of atomic position (RMSD) of 0.843 Å, which indicates close similarity. 

The superimposed structure of the AlphaFold2 model and the bound-form homology model differed in the loop position (Figure 3B). The overall RMSD was 0.932 Å. In the AlphaFold2 structure and in the apo-protein form, the loop crosses the ligand, preventing binding of the substrate to the active site. 

### 2.4. Enzyme Kinetics

The aforementioned amino-acid exchanges of identified critical residues (C707A: Cys → Ala; E673D: Glu → Asp; E673A: Glu → Ala) were experimentally validated by means of site-directed mutagenesis of 10-formyl-THFDH of *A. mellifera*. All recombinant derivatives were successfully expressed and purified. The dehydrogenase activity was measured by monitoring the increase in absorbance at 340 nm. The recombinantly expressed wildtype protein showed an enzyme activity with a Vmax of 0.1846 nM·min^−1^ to 0.4253 nM·min^−1^ at different pH and Km values of 2.455 µM up to 9.596 µM (according to previously published data [22]). The mutations lost their activity almost entirely, with values ranging between 0.007 nM·min^−1^ and 0.12 nM·min^−1^ for their Vmax values and between 8.33 × 10^−14^ µM and 8.57 × 10^−16^ µM for their Km values; additionally, it has to be stated that all R^2^ values, indicating the goodness of fit, for the mutants were below 0.17, which indicates a poor fit of the used nonlinear regression (Figure 4, Table 1). 

## 3. Discussion

Natural colonization of floral nectar by yeast and bacteria [36,37,38,39] often results in the production of methanol and ethanol by fermentation [40,41]. Hence, bees need to be able to detoxify formic acid, as it is the main driver of methanol toxicity [42]. The strong conservation of the enzyme within the family of *Apidae* (Appendix A) underlines the importance of the protein characterized here, which exhibits 100% conservation of the active residues E673 and C707 and the surrounding region, even in the relatively distinct genera *Eufriesea* and *Apis* [43] and between Mammalia and *Apidae*.

The folate-dependent one-carbon pool (C1) is known to be the major detoxification pathway of formic acid in mammals and presumably also in *A. mellifera*, catalyzing the conversion of THF to 10-formyl-THF by a 10-formyl-THF synthase [24]. Subsequently, the aforementioned 10-formyl-THFDH catalyzes the NADP^+^-dependent reaction of 10-formyl-THF to CO_2_ and THF [25,26]. This highlights the importance of folic acid as one of the main factors influencing the efficiency of this detoxification mechanism, as it is the first substrate in the reaction chain. A dietary supplementation might be beneficial for the honey bee to optimize survival during formic acid treatments. On the basis of our molecular data, further research in this area is now being encouraged, e.g., to better understand the detoxification of formic acid used against *V. destructor* in honey bees in the field. The next aim is to use this knowledge for developing better and more bee-friendly formic acid treatment strategies. 

AlphaFold2 is an incredible advance in the protein modeling world. This study shows how close AlphaFold2 structural predictions come to the more work-intense conservative methods, such as homology modeling. The difference between the apo-form enzyme and the AlphaFold2 model is almost negligible with an RMSD of 0.843 Å. With a difference of 0.932 Å between the bound-form homology model and the AlphaFold2 prediction, the model also appears to be well suited for docking models. This is a misconception, because the main difference in the RMSD is in the most important part, the loop that blocks the substrate entrance tunnel in the apo-form (Figure 2A). This cannot be correctly predicted, as AlphaFold2 is unable to include co-factor binding in its prediction (Figure 3B). Additionally, the “dark proteome”, which is estimated to comprise 44–54% of all proteins in eukaryotic cells, consists of proteins that are unstably folded and, therefore, have no well-defined three-dimensional structure [44,45]. Such proteins are thought to play a role in defense or signal transduction and change their structure when interacting with different macromolecules such as RNA and other proteins. Another example where AlphaFold2 is currently reaching its limits are enzymes with co-factors. In these enzymes, a conformational change is initiated, which in turn enables the enzyme to bind its specific substrate to trigger further reactions [28]. AlphaFold2 seems to be unable to predict loop changes caused by the binding of before mentioned cofactors. This can lead to problems in the prediction of binding sites, as shown in our example. 

Despite the low conservation of the enzyme 10-formyl-THFDH between mammals and *A. mellifera,* this study shows that the active site of the dehydrogenase subunit remains almost unchanged. On one hand, Tsybovsky and Krupenko [46] have already shown that the residues C707 and E677 are important for the degradation of 10-formyl tetrahydrofolate in *Rattus norvegicus*. However, on the other hand, the conservation of this enzyme between the two species *A. mellifera* and *R. norvegicus* is so low [22] that confirmation of the active site is necessary to make further functional statements. The expected molecular mode of action is as follows: first, glutamate E673 and cysteine C707 are hydrogen-bonded. The binding of NADP^+^ results in the rotation of the glutamate sidechain away from the cysteine; thereafter, the negatively charged sulfur of the cysteine forms a transient covalent bond with the C4 atom of the nicotinamide ring of the NADP^+^. In the two phases of the dehydrogenase catalysis, acetylation and deacetylation, the cysteine fulfills a nucleophilic catalytic function, whereas the glutamate is expected to activate a water molecule in the deacetylation step [46]. With the proposed mechanism, the two mentioned residues are of great importance. The mutations showed no activity in comparison to the wildtype enzyme. Additionally, the homology modeling and docking revealed one main problem with the binding site of the substrate, when one of the key residues was changed. The steric clash between the substrate and F870 of the enzyme seemed to result in almost no activity in all mutants investigated (Appendix A). A double mutation of the wildtype, involving a mutation of the phenylalanine and the E673 residue of the enzyme, could shed light on whether this steric clash is the reason why the glutamic acid for aspartic acid exchange (E673D) loses its activity completely, although only a decrease in activity is to be expected, since aspartic acid and glutamic acid are similar in their biochemical properties.

In conclusion, our approach allowed verification of the active site of honeybee 10-formyl-THFDH, although only evolutionary very distant enzymes have been experimentally confirmed so far. In addition, the importance and detoxifying potential of the honey bee 10-formyl-THFDH was demonstrated at the molecular level, which would also be expected for other species in the family of *Apidae*. At the same time, this study points to certain limitations of the currently used AlphaFold2 algorithm.

## 4. Materials and Methods

### 4.1. Homology Modeling

The homology models were built by Prime [47]. The ligand-bound form loop of the homology model was built by Prime (Schrödinger Release 2019-4).

### 4.2. Covalent Docking

The homology model was used to generate the receptor grid. The substrate was prepared by LigPrep. The force field was OPLS3 [48]. The covalent docking method from Glide was used to dock the substrate into the receptor [49].

### 4.3. AlphaFold2 Structural Prediction

The full-length 10-formyl-THFDH predicted structure was obtained by AlphaFold2 Python code on Google Colab (Appendix A) (https://colab.research.google.com/github/deepmind/alphafold/blob/main/notebooks/AlphaFold.ipynb, accessed on 14 April 2022) [27].

### 4.4. Structural Analysis

The electrostatic surface of the binding pocket was analyzed by ChimeraX. The RMSD values between two different structures were calculated by PyMOL (Version 2.4.2 Schrödinger, LLC.) [50,51].

### 4.5. Visualization

All modeling figures were processed and presented by ChimeraX [50,51].

### 4.6. Multiple Sequence Alignment

ClustalΩ [35] was used for amino-acid sequence alignment. Thirteen consensus sequences of different species of the family *Apidae* from across the globe and selected mammalian species were used (Appendix A).

### 4.7. A. Mellifera Sampling

One day old worker bees were collected from the apiary of the Institute of Veterinary Biochemistry, Freie Universität Berlin, Berlin (52.42898° N, 13.23762° E) using one queen-right colony with *A. mellifera* (carnica) in the summer season 2020. Colonies were healthy, had enough food supply, and showed no symptoms of diseases or increased parasitism. Individuals were shock-frozen in liquid nitrogen and stored at −80 °C until further use.

### 4.8. RNA Extraction

RNA extraction was performed using the Quick-RNA™ Miniprep Kit (Zymo Research Europe GmbH, Freiburg, Germany). Briefly, individuals were lysed in a lysing Matrix S (MP Biomedicals, Heidelberg, Germany) containing 1 mL of lysis buffer using a BeadBlaster (Benchmark Scientific, Edison, NJ, USA). Tubes were then centrifuged at 12,000× *g* at 4 °C for 10 min. The supernatant was transferred into a clean microcentrifuge tube containing 1× volume of 100% ethanol. The solution was then used according to manufacturer’s protocol. RNA was eluted in a total volume of 50 µL of ddH_2_O. The quantity and quality of total RNA were analyzed using an Agilent RNA 6000 nano chip on a 2100 Bioanalyzer (Agilent Technologies, Santa Clara, CA, USA). Isolated RNA was stored at −80 °C until use.

### 4.9. First-Strand cDNA Synthesis

Protoscript^®^ II Transcriptase (New England Biolabs, Inc., Ipswich, MA, USA) was used according to the manufacturer’s protocol. Briefly, 1 µg of DNA-free RNA was incubated with 1 µL of d(T)23VN-Primer (50 µM) and 1 µL of Random Primer Mix (50 µM) at 65 °C for 5 min in a total volume of 8 µL. Thereafter, 12 µL of Protoscript Mastermix was added, and the sample was incubated at 42 °C for 60 min and heat-inactivated at 80 °C for 5 min. The cDNA was then diluted by the addition of 80 µL of ddH_2_O and stored at −20 °C in adequate aliquots. To create a broad library, 5 µL of each sample was added to one microcentrifuge tube before freezing.

### 4.10. Creation of pFBD-eGFP-Amel_10-Formyl-THFDH Expression Vector

The open reading frame of the wildtype *A. mellifera* cytosolic 10-formyl-THFDH (Accession: XM_026442355.1) (Amel_10-formyl-THFDH) was amplified by polymerase chain reaction (PCR) using the primers Amel_FTHFDH_ORF_F/R along with the full-length cDNA (Table 2). The PCR product was then subcloned into pJet1.2 vector (Thermo Scientific, Karlsruhe, Germany) for sequencing and creation of a template for further use. The ORF-containing vector was used to create overhangs containing restriction sites (BamHI, NotI) and a 6×-HisTag at the N-terminus for later purification of the protein. The pFastBacDual (pFBD) vector of the Bac-to-Bac System (Thermo Scientific, Karlsruhe, Germany) with an enhanced green fluorescent protein (eGFP) cloned at the p10-promoter site was used as expression vector. The insert was created by PCR using the Amel_FTHFDH_BHI_HT_F and Amel_FTHFDH_NotI_R primers (Table 2). pFBD-eGFP was digested with appropriate restriction enzymes, and the vector was dephosphorylated using an Antarctic Phosphatase (New England Biolabs, Inc., Ipswich, USA) to prevent religation. The PCR product was ligated with the vector using a T4-ligase (New England Biolabs, Inc., Ipswich, USA) according to standard protocols.

### 4.11. Mutagenesis of the Expression Vector

For the mutagenesis of the expression vector and creating the three mutants (C707A, E673D, and E673A), two different approaches were used. Firstly, the KLD enzyme mix from New England Biolabs was used to create the expression vectors E673D and E673A according to the manufacturer’s protocol using the primers pFBD_Amel_FTHFDH_E673A_F/R and pFBD_Amel_FTHFDH_E673D_F/R (Table 2). For the creation of the expression vector for the mutant C707A, an assembly PCR approach was used [52], due to the KLD mix not working for this mutant. Briefly, Amel_FTHFDH_C707A_Ass-F/R, Amel_FTHFDH_BHI_HT_F, and Amel_FTHFDH_NotI_R primer (Table 2) were used to create partly overlapping amplicons, which were thereafter assembled to create the full-length ORF with the appropriate mutation. Site-directed mutagenesis was performed using 50 ng of purified template and a Q5-Polymerae (NEB). After the initial step at 98 °C for 5 min, the first 10 cycles were run for 30 s at 98 °C and 20 s at 56 °C + dT of 0.5 °C per cycle, followed by 1.5 min at 72 °C. The next 30 cycles were conducted according to the following scheme: 30 s at 98 °C, 20 s at 61 °C, and 1.5 min at 72 °C, followed by a terminal step at 72 °C for 3 min. Amplicon assembly was performed using a two-step protocol. First, equimolar amounts of both gel purified amplicons were added to the reaction containing 1 U of Q5-Polymerase, dNTPs, and 10× buffer in 25 µL of total volume. The assembly was started at 98 °C for 5 min, followed by 10 cycles with 30 s at 98 °C, 20 s at 61 °C, and 1.5 min at 72 °C. The reaction was cooled down to 4 °C, and primers Amel_FTHFDH_BHI_HT_F and Amel_FTHFDH_NotI_R were added. Thereafter, the amplification was performed using the following protocol: 98 °C for 5 min, followed by 30 cycles of 98 °C for 30 s, 61 °C for 20 s, and 72 °C for 2 min, followed by a terminal step at 72 °C for 5 min. The mutated insert was inserted to the expression vector as described under Section 2.4.

### 4.12. Creation of the Recombinant Bacmid

To create the recombinant Bacmid, Gibco™ Max Efficiency™ DH10Bac competent cells (Thermo Scientific, Karlsruhe, Germany) were transformed using 1 µg of pFBD construct. The cells were thawed on ice, and 1 µg of construct was added. The mixture was incubated for 30 min on ice, heat-shocked for 45 s at 42 °C, and transferred back to ice for 2 min. Then, 900 µL of SOC medium was added. The culture was incubated for 4 h at 37 °C in a shaking incubator at 225 rpm. The cells were plated on LB medium containing 50 µg/mL kanamycin, 7 µg/mL gentamicin, 10µg/mL tetracycline, 500 µg/mL X-Gal. and 1 µM IPTG. The plates were incubated for 48 h at 37 °C. White colonies were restreaked, and Bacmid was isolated according to the manufacturer’s protocol.

### 4.13. Creation of Baculovirus

To create recombinant baculovirus, Sf21 insect cells (Thermo Scientific, Karlsruhe, Germany) were transfected with 1 µg of Bacmid DNA. Then, 6 µL of Gibco™ Cellfectin™ II reagent (Thermo Scientific, Karlsruhe, Germany) was used as suggested by the manufacturer. Successful transfection was monitored by expression of eGFP under an inverse fluorescent microscope DMI 6000B (Leica), and photos were taken using a DFC 365FX (Leica) camera. Virus stock was extracted by detaching cells from flask and centrifuging at 3000× *g* for 5 min. Virus containing supernatant was transferred into a sterile 15 mL centrifuge tube and stored safe from light at 4 °C until further use.

### 4.14. Expression and Purification of Recombinant Proteins

To produce recombinant protein, Hi5 cells (Thermo Scientific, Karlsruhe, Germany) at 80–90% confluency were used. A total of 3 × 10^6^ cells were seeded into a T175 flask (Sarstedt) containing 50 mL of Gibco™ ExpressFive™ SFM (Thermo Scientific, Karlsruhe, Germany). Then, 30 µL/mL Virus-Stock was added, and cells were incubated at 27 °C for 4 days or until most cells showed eGFP expression. Cells were pelleted at 5000× *g* for 20 min at 4 °C. Pellets were resuspended in 20 mL of PBS containing EDTA-free proteinase inhibitor cocktail (SIGMA-ALDRICH, Darmstadt, Germany). The suspension was sonified on ice using a sonifier 250 (Branson) for 4 min with an amplitude of 2 at 20% energy. The suspension was cleared by centrifugation at 5000× *g* for 20 min at 4 °C. Protein-containing supernatant and PBS containing imidazole at different concentrations (10 mM (equilibration buffer); 25 mM (wash buffer); 100 mM, 150 mM, 200 mM, and 500 mM (elution buffer)) were particle-free filtered (0.45 µM pore size, PES). Next, 2 mL bed-volume (BV) of HisPur™ Ni-NTA Resin (Thermo Scientific, Karlsruhe, Germany) was equilibrated with 5 BV of equilibration buffer. The protein was equilibrated with 20 mL of equilibration buffer and added to the column. The column was washed with 20 BV of wash buffer; thereafter, four elution fractions were obtained using four different concentrations of imidazole (100 mM, 150 mM, 200 mM, and 500 mM). The whole purification was performed at 4 °C. Thereafter, to remove impurities and imidazole from the enzyme, protein concentrators with a molecular weight cutoff of 50 kDa (Pierce) were used as suggested by the manufacturer.

### 4.15. Synthesis of 10-Formyl-THF

To synthesize the substrate 10-formyl-THF, an established protocol by Rabinowitz et al. [53] was used. Briefly, 100 mg of dl-5-formyltetrahydrofolic acid (SIGMA-ALDRICH, Darmstadt, Germany) was dissolved in 8 mL of 1 M β-mercapto-ethanol (Roth). The pH was adjusted to 1.5 with HCl. The mixture was stored at 4 °C for at least 12 h. The solution containing dl-5,10-methenyltetrahydrofolic acid as a precipitate (bright yellow tint), was adjusted to pH 8 with KOH, purged with argon, and incubated overnight at 4 °C in an evacuated vessel. The final solution containing 10-formyl-THF (clear color) was directly used for enzyme assays.

### 4.16. Enzyme Activity Assays

All assays were performed using a ClarioStar plus multimode plate reader (BMG labtech). All mutants were tested with the same batch of synthesized substrate. First, 100 mM β-mercapto-ethanol, 200 µM NADP^+^, and 10 µg of purified enzyme were added to each well and incubated at 30 °C for 2 min. The substrate was injected using built-in injectors at different concentrations. NADPH production was monitored at 340 nm for a period of 30 min. All substances were diluted in Tris/HCl buffer (pH 6.8–8.4) to a total of 100 µL. The K_m_ and V_max_ were calculated using a molar extinction coefficient of 6220 M^−1^·cm^−1^ for NADPH.

### 4.17. Analysis of Kinetic Data

Initial reaction rates were used to determine the respective enzyme activities. Kinetic parameters were derived using GraphPad Prism version 9.0.2 (for Windows 10, GraphPad Software, San Diego, CA, USA, www.graphpad.com, accessed on 3 March 2022), which determined the kinetic parameters from the Michaelis–Menten equation using nonlinear regression.

## Figures and Tables

**Figure 1 ijms-24-00354-f001:**
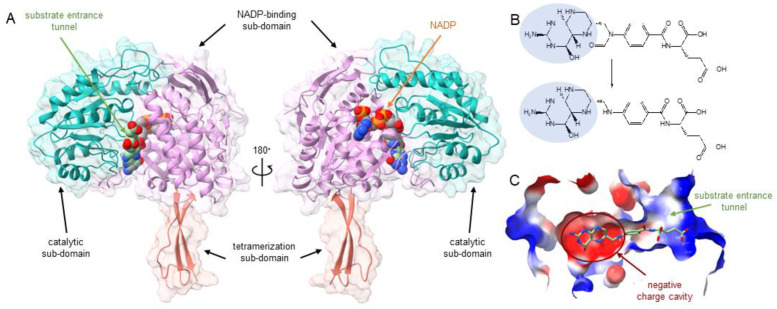
(**A**) The C-terminal structure of 10-formyl-THFDH. The C-terminal 10-formyl-THFDH domain includes the catalytic subdomain (turquoise), the NADP^+^-binding subdomain (light pink), and the tetramerization subdomain (salmon). On the left is the substrate entrance tunnel. The substrate (10-formyl-THF) is shown in light green. The NADP^+^-binding site is opposite to the substrate-binding site. On the right is the structure rotated 180°. NADP^+^ is shown in gray, Oxygen atoms are shown in red, and nitrogen atoms are shown in blue. The substrate carbon atoms are shown in light green. NADP^+^ carbon atoms are shown in gray. (**B**) The substrate reaction. The highlighted areas of substrate and product show the positively charged region (for pKa-calculation see Appendix A). After the enzymatic reaction, the substrate is oxidated, and one molecule of CO_2_ (not shown) is produced. (**C**) Clip of the substrate entrance tunnel with its negatively charged cavity to bind the substrate.

**Figure 2 ijms-24-00354-f002:**
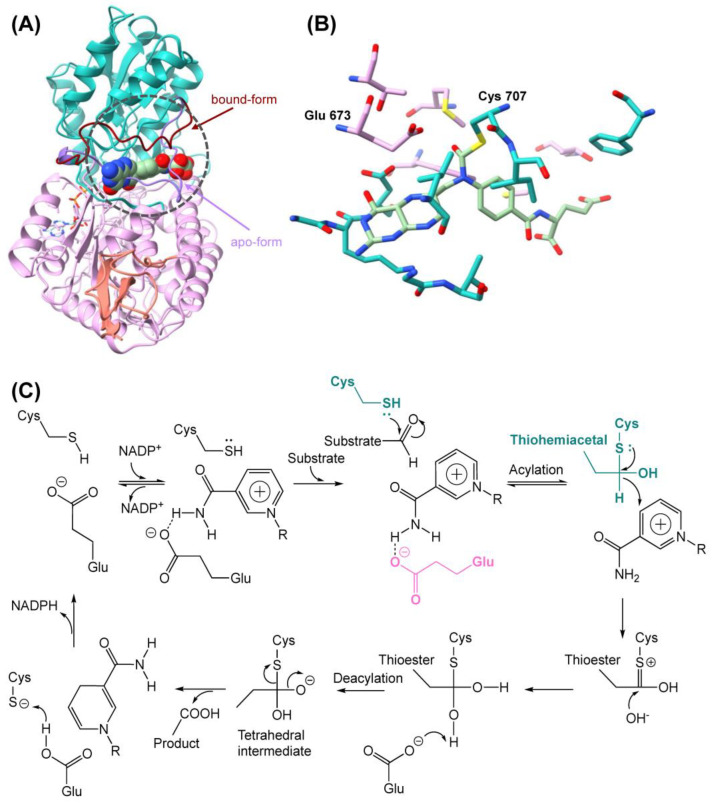
(**A**) The superimposed structure of apo-form and binding-form of 10-formyl-THFDH. The light purple loop is the apo-form. The red loop is the bound form. The substrate molecule is shown in ball representation. Substrate carbon atoms are shown in light green, oxygen atoms are shown in red, and nitrogen atoms are shown in blue. The apo-form loop shown in purple has a steric clash with the substrate molecule. Once the enzyme is activated by NADP^+^, the loop is fully opened and forms the substrate entrance tunnel, and the substrate enters the binding pocket. (**B**) Zoomed-in view of the binding pocket of the substrate, which is covalently bound to the C707. (**C**) The 10-formyl-THFDH oxidation reaction mechanism. The lone pair of electrons of the sulfur in the thiol group of the cysteine nucleophilically attacks the oxygen from formyl group of the substrates.

**Figure 3 ijms-24-00354-f003:**
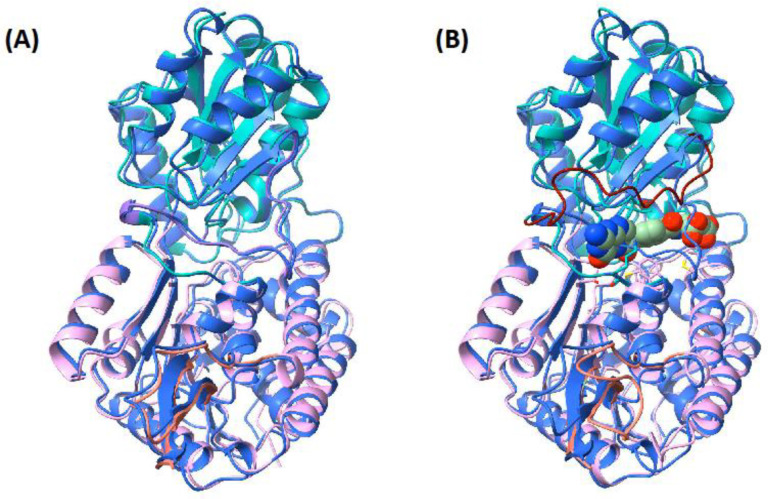
The superimposed models of the homology model and the AlphaFold2 model. (**A**) The AlphaFold2 model (blue) and the apo-form of the enzyme (teal: catalytic subdomain; pink: NADP^+^-binding subdomain; orange: tetramerization subdomain). The RMSD of these two structures is 0.843 Å. The AlphaFold2 structure shows a similar loop to the apo-form homology model. (**B**) The superimposed view of the AlphaFold2 structure and the ligand-bound form of the homology model (teal: catalytic subdomain; pink: NADP^+^-binding subdomain; orange: tetramerization subdomain; dark red: open loop). The RMSD of these two structures is 0.932 Å. The AlphaFold2 structure shows that the loop clashes with the ligand.

**Figure 4 ijms-24-00354-f004:**
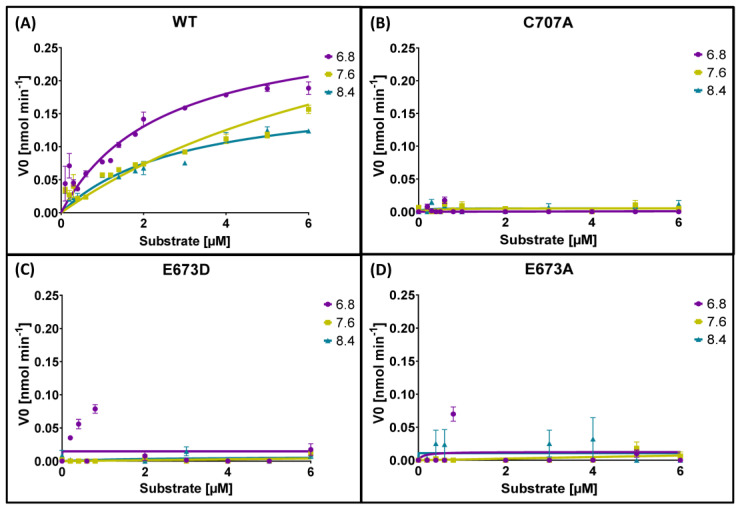
(**A**) Activity of the recombinantly expressed wildtype enzyme 10-formyl-THFDH. (**B**) Enzyme activity of the mutation with an alanine exchange at the cysteine 707 position (C707A). (**C**) Enzyme activity of the mutation with an aspartic acid exchange at the glutamic acid 673 position (E673D). (**D**) Enzyme activity of the mutation with an alanine exchange at the glutamic acid 673 position (E673A).

**Table 1 ijms-24-00354-t001:** Vmax and Km values with statistics. The calculated Vmax and Km values for each mutant 10-formyl-THFDH (including wildtype (WT)), as well as their respective 95% confidence interval and the goodness of fit as R^2^.

pH	Enzyme	V_max_	K_m_	95% CI V_max_	95% CI K_m_	R^2^
6.8	WT	0.2907	2.455	0.2515 to 3.542	1.72 to 3.542	0.8290
	C707A	NA	8.57 × 10^16^	NA	NA	−0.1333
	E673A	0.01265	0.1109	0.0026 to 17,623,525	NA	0.0207
	E673D	0.01484	8.33 × 10^−14^	0.0078 to 0.02894	NA to 0.2658	0.0262
7.6	WT	0.4253	9.596	0.2908 to 83.76	5.133 to 23.98	0.8858
	C707A	0.00538	0.228	0.0023 to 0.01327	NA to 4.951	−0.0632
	E673A	0.1299	102.8	0.0037 to NA	0.2755 to NA	0.1713
	E673D	0.05799	81.54	0.0021 to NA	NA	0.1493
8.4	WT	0.1846	2.961	0.1520 to 0.2346	1.934 to 4.670	0.9029
	C707A	0.005269	0.09273	0.0025 to 26,122,669	NA	0.0408
	E673A	0.0108	1.54 × 10^−10^	0.0017 to 9,515,687	NA	0.01661
	E673D	0.006758	2.156	0.0018 to 1,634,800,901	NA	0.0050

**Table 2 ijms-24-00354-t002:** Primer used for vector creation and mutagenesis.

Name	5’–3’ Sequence
Amel_FTHFDH_ORF_F	ATGGCGCAACTCAAAGTGGC
Amel_ FTHFDH _ORF_R	CTAATATTCTACAGTGATAGTTTTTG
Amel_ FTHFDH _BHI_HT_F	TCATACGGATCCATGCACCACCACCACCACCACGCGCAACTCAAAGTGGC
Amel_ FTHFDH _NotI_R	TCATACGCGGCCGCCTAATATTCTACAGTGATAGTTTTTG
Amel_ FTHFDH _E673A_F	ATCCCTAGCATTAGGTGGAA
Amel_ FTHFDH _E673D_F	ATCCCTAGACTTAGGTGGAA
Amel_ FTHFDH _E673AD_R	ACTTTCTTCAAATTACTATT
Amel_ FTHFDH _C707A_Ass_F	CAAAGGAGAAAACGCAATAG
Amel_ FTHFDH _C707A_Ass_R	CTATTGCGTTTTCTCCTTTGTTGAAGAACAC

## Data Availability

All data are shown in the publication.

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
