# Peer review of "The Active Site of the Enzyme 10-Formyl-THFDH in the Honey Bee *Apis mellifera*—A Key Player in Formic Acid Detoxification"

_ijms, 2022, doi:10.3390/ijms24010354_

Round 1

Reviewer 1 Report

The active sites of the enzyme 10-formyl-THFDH in honey bee were characterized by combining homology modeling and protein-mutagenesis. Additionally, this study points to certain limitations of the currently used AlphaFold2 algorithm. Results strongly support the conclusion that the residues C707 and E677 are the active site in the10-formyl-THFDH enzyme. It is a very interesting research.

There are some comments on this article.

Line11 “the them”

Line29,30,32 Varroa destructor should be italic; please check full text for these Latin names

Line 136, Line281, and Line288, subtitle,  2.2..  so many subtitles have two dot.

line175 and 176 min-1

Line 178, 8.33 x10-14 µ M, not x, it should be ×

Line 178-179 and the corresponding data, 8.57 ×1016 µM, is it correct? Maybe it is 8.57 ×10-16 µM?

Line 282, 1-day-old worker is better;

Line 292, 293, not x, it should be ×

Line372, 5.000 x g, should be 5 000×g or 5,000×g, not dot

This work will help us to protect honey bees from the damage of other pests. I am very interested in this research. I have a question not related to these research results. we know that many hymenopteran insect including honey bee store formic acid in glands and secrete formic acid to defense. What is the situation in these glands? Upregulation of 10-formyl-THFDH? Or some other special way to protect themselves from formic acid.

Author Response

The active sites of the enzyme 10-formyl-THFDH in honey bee were characterized by combining homology modeling and protein-mutagenesis. Additionally, this study points to certain limitations of the currently used AlphaFold2 algorithm. Results strongly support the conclusion that the residues C707 and E677 are the active site in the10-formyl-THFDH enzyme. It is a very interesting research. 

There are some comments on this article.

Line11 “the them”

Answer: Was improved accordingly

Line29,30,32 Varroa destructor should be italic; please check full text for these Latin names

Answer: Was improved accordingly

Line 136, Line281, and Line288, subtitle,  2.2..  so many subtitles have two dot.

Answer: Was improved accordingly

line175 and 176 min-1

Answer: Was improved accordingly

Line 178, 8.33 x10-14 µ M, not x, it should be ×

Answer: Was improved accordingly

Line 178-179 and the corresponding data, 8.57 ×1016 µM, is it correct? Maybe it is 8.57 ×10-16µM?

Answer: Was improved accordingly

Line 282, 1-day-old worker is better;

Answer: Was improved accordingly

Line 292, 293, not x, it should be ×

Answer: Was improved accordingly

Line372, 5.000 x g, should be 5 000×g or 5,000×g, not dot

Answer: Was improved accordingly

This work will help us to protect honey bees from the damage of other pests. I am very interested in this research. I have a question not related to these research results. we know that many hymenopteran insect including honey bee store formic acid in glands and secrete formic acid to defense. What is the situation in these glands? Upregulation of 10-formyl-THFDH? Or some other special way to protect themselves from formic acid.

Comment: This is a very interesting question - we don't know anything about this yet, but it would be a new research project.

Reviewer 2 Report

My comments:

This manuscript deals with formic acid detoxification in honey bees by using bee enzyme, 10-formyl-THFDH, which can trigger the removal of formic acid from the organism. This study covers the characterization of active sites of this enzyme by combining homology modeling, protein-mutagenesis, and the 3D protein structure which could provide its molecular docking characteristics and detoxification mechanisms of formic acid in Apis mellifera.

This manuscript is well written and provides deep analysis, thus it should be accepted for publication in International Journal of Molecular Sciences after few minor corrections/suggestions, as follows.

1.      Page 1, line 11; It should be “it weakens them by …”

2.      Page 1, line 27; It should be “Various factors, such as …”

3.      Page 1, line 27; It should be “Owing to …”

4.   Page 3; Caption of Figure 1 should be corrected. Basically, the figure caption just the title of the image, graph, or picture. So, a full sentence is not necessary. For instance, Figure 1. A) the C-terminal structure of 10-formyl-THFDH. … (B) The substrate reaction. …. (C) Clip of the substrate entrance tunnel with its negatively charged cavity to bind the substrate.

5.      Page 4; Caption of Figure 2 should be corrected. See Comment #4.

6.      Page 5; Caption of Figure 3 should be corrected. See Comment #4.

7.  Page 5, line 175; It should be “Vmax of 0.1846 nM min-1”; (-1 is superscript). Please check throughout the manuscript.

8.      Page 6; Figure 4. The data points in the figure are too small, so authors should change them to a bigger size.

9.      Page 6, line 188; the title of Table 1; delete “This table shows”.

Author Response

This manuscript deals with formic acid detoxification in honey bees by using bee enzyme, 10-formyl-THFDH, which can trigger the removal of formic acid from the organism. This study covers the characterization of active sites of this enzyme by combining homology modeling, protein-mutagenesis, and the 3D protein structure which could provide its molecular docking characteristics and detoxification mechanisms of formic acid in Apis mellifera.

This manuscript is well written and provides deep analysis, thus it should be accepted for publication in International Journal of Molecular Sciences after few minor corrections/suggestions, as follows.

  1. Page 1, line 11; It should be “it weakens them by …”

Answer: Was improved accordingly

  1. Page 1, line 27; It should be “Various factors, such as …”

Answer: Was improved accordingly

  1. Page 1, line 27; It should be “Owing to …”

Answer: Was improved accordingly

  1. Page 3; Caption of Figure 1 should be corrected. Basically, the figure caption just the title of the image, graph, or picture. So, a full sentence is not necessary. For instance, Figure 1. A) the C-terminal structure of 10-formyl-THFDH. … (B) The substrate reaction. …. (C) Clip of the substrate entrance tunnel with its negatively charged cavity to bind the substrate.

Answer: Was improved accordingly

  1. Page 4; Caption of Figure 2 should be corrected. See Comment #4.

Answer: Was improved accordingly

  1. Page 5; Caption of Figure 3 should be corrected. See Comment #4.

Answer: Was improved accordingly

  1. Page 5, line 175; It should be “Vmax of 0.1846 nM min-1”; (-1 is superscript). Please check throughout the manuscript.

Answer: Was improved accordingly

  1. Page 6; Figure 4. The data points in the figure are too small, so authors should change them to a bigger size.

Answer: A suitably modified figure was put

  1. Page 6, line 188; the title of Table 1; delete “This table shows”.

Answer: Was improved accordingly